# GRAPH-PDE: COUPLED ODE STRUCTURE FOR GRAPH NEURAL NETWORKS

## ABSTRACT

Spike Neural Networks (SNNs), a type of ordinary differential equation (ODE), have evolved as a sophisticated approach for addressing challenges in dynamic graph neural networks. They typically sample binary features at each time step and propagate them using SNNs to achieve spike representation. However, spike ODE remain discrete solutions, limiting their ability to capture continuous changes and subtle dynamics in time series. An effective solution is to incorporate continuous graph ODE into spike ODE to model dynamic graphs from two distinct dimensions, i.e., the time dimension of graph ODE and the latency dimension of spike ODE. The key challenge is to design a structure that seamlessly integrates spike ODE and graph ODE while ensuring the stability of the model. In this paper, we propose Graph-PDE (G-PDE), which combines spike and graph ODE in a unified graph partial differential equation (PDE). Considering the incorporation of high-order structure would preserve more information, alleviating the issue of information loss in first-order ODE. Therefore, we derive the high-order spike representation and propose the second-order G-PDE. Additionally, we prove that G-PDE addresses the issues of exploding and vanishing gradients, making it easier to train deep multi-layer graph neural networks. Finally, we demonstrate the competitive performance of G-PDE compared to state-of-the-art methods across various graph-based learning tasks.

## 1 INTRODUCTION

Spike Neural Networks Ghosh-Dastidar & Adeli (2009a;b) (SNNs) are a type of artificial neural network that mimic the behavior of biological neurons, enabling them to process information in a manner similar to the human brain. Unlike traditional neural networks, which operate based on continuous activation values, SNNs utilize discrete spikes or pulses of activity for representation. The unique characteristic of SNNs lies in their ability to capture temporal dynamics and process time-varying information. SNNs are typically applied in tasks that involve event-based processing Yao et al. (2021), such as sensory perception Kumarasinghe et al. (2021); Rapp & Nawrot (2020), real-time data analysis Zhu et al. (2020b); Bauer et al. (2019), and graph machine learning Li et al. (2023); Xu et al. (2021); Zhu et al. (2022).

While the application of SNN to graph-based time series prediction problems is increasingly diverse, these approaches typically involve sampling the graph at each time step and employing discrete forward propagation methods for training. However, such methods are unable to handle irregularly sampled observations and require access to all node observations at every timestamp Huang et al. (2020; 2021). In contrast, graph ordinary differential equation (ODE) techniques excel in capturing the continuous changes and subtle dynamics of graphs Poli et al. (2019); Huang et al. (2020; 2021); Gupta et al. (2022). Specifically, they leverage GNNs initially for state representation initialization and subsequently develop a neural ODE model that governs the evolution of both nodes and edges within the dynamical system. Finally, a decoder is employed to generate predictions while optimizing the entire generative model through variational inference.

However, combining the SNN and graph ODE for continuous graph learning is difficult due to the following challenges. First, *how to incorporate the SNN and graph ODE into a unified model?* SNN and graph ODE respectively integrate information along two different time dimensions, i.e., the latency dimension in SNN and the time dimension in graphs. How to incorporate the dual process

into a unified framework to preserve the advantages of low-power consumption from SNN and subtle dynamics from graph ODE is a challenge. Second, *how to alleviate the problem of information loss of SNN?* SNN achieves the requirement of low-power consumption by binarizing continuous features, but a large amount of detailed information will be lost during the binarization process, resulting in poor performance. Therefore, how to alleviate the problem of information loss in SNN is another challenge. Third, *how to guarantee the stability of the proposed method?* Traditional graph methods face the exploding and vanishing gradients problem for training deep multi-layer GNNs. Therefore, how to design a stable model for graph learning is the third challenge.

In this paper, we identify SNNs as a form of ODE known as spike ODE and introduce Graph-PDE (G-PDE), a novel approach that combines spike and graph ODEs within a unified graph partial differential equation (PDE) framework. Considering the benefits of high-order structures to preserve more information and overcome information loss in first-order ODEs, we derive the high-order spike representation and introduce the second-order G-PDE. Additionally, we demonstrate that G-PDE effectively tackles challenges related to exploding and vanishing gradients. This property facilitates the training of deep multi-layer graph neural networks. We conduct comprehensive evaluations of G-PDE against state-of-the-art methods across various graph-based learning tasks, showcasing its effectiveness and broad applicability in the field. The contributions can be summarized as follows:

- Novel Architecture. Our approach incorporates the coupled ODE (spike ODE and graph ODE) into a unified graph PDE, which reserves the advantages of low-power consumption from spike ODE and extraction of continuous changes and subtle dynamics from graph ODE. In addition, we propose the second-order graph PDE, which would alleviate the information loss issue in the first-order PDE.

- Second-order Spike Representation. We first derive the second-order spike representation and study the backpropagation of second-order spike ODE, which is further applied in the second-order graph PDE.

- Theoretical Analysis of the Gradient. To guarantee the stability of the proposed G-PDE, we prove that G-PDE mitigates the problem of exploding and vanishing gradients and improves the trainingability of deep multilayer GNNs.

- Extensive Experiments. We evaluate the proposed G-PDE on extensive datasets on a variety of graph learning tasks. The results show that our proposed G-PDE outperforms the variety of state-of-the-art methods.

## 2 RELATED WORK

### 2.1 SPIKING NEURAL NETWORK

SNNs have become a promising solution in solving the graph machine learning problems. Currently, there are two primary directions of SNNs. The first is to establish a connection between SNNs and Artificial Neural Networks (ANNs), which enables the ANN-to-SNN conversion Diehl et al. (2015); Hunsberger & Eliasmith (2015); Rueckauer et al. (2017); Rathi et al. (2020), and the optimization of SNNs using gradients computed from these equivalent mappings Thiele et al. (2020); Wu et al. (2021); Zhou et al. (2021); Xiao et al. (2021); Meng et al. (2022b). However, these methods usually require a relatively large number of time-steps to achieve performance comparable to ANNs, suffering from high latency and usually more energy consumption. The second direction is to directly train SNNs with backpropagation Bohte et al. (2000); Esser et al. (2015); Bellec et al. (2018); Huh & Sejnowski (2018), which typically employs the surrogate gradients Shrestha & Orchard (2018) method to overcome the non-differentiable nature of the binary spiking. This follows the backpropagation through time (BPTT) framework. BPTT with surrogate gradients can achieve extremely low latency, however, it requires large training memory to maintain the computational graph unfolded over time. However, there are few works that focus on the dynamic graph of spikes, and the merely work Li et al. (2023) simply applies the SNNs into the dynamic graph, which cannot capture the continuous changes and subtle dynamics in time series. Our work combines the SNN with the graph ODE ingeniously, which reserves the advantages of low-power consumption from SNN and extraction of continuous changes and subtle dynamics from graph ODE.

## 2.2 Graph ODE

Recently, several ODE methods have been developed Chen et al. (2018); Dupont et al. (2019) to model continuous depth by parameterizing the derivative of the hidden state. To enhance the expressive capability of graph models, many works have integrated neural ODEs into graph neural networks (GNNs) Xhonneux et al. (2020); Rusch et al. (2022); Qin et al. (2023); Luo et al. (2023). However, these methods primarily focus on designing continuous models to address over-smoothing issues encountered in static graphs. In our work, we extend the concept of graph ODEs to graph PDEs by incorporating an additional dimension, i.e., the latency dimension in spike ODEs. In this way, we propose a flexible framework that enables continuous graph learning while preserving the advantages of both spike and graph ODEs simultaneously.

## 3 Preliminaries

### 3.1 Spiking Neural Networks

**First-order SNNs**. Different from traditional artificial neural networks, a spiking neural network (SNN) samples input data into binary spikes over time, and each neuron in the SNN maintains a membrane potential that integrates input spikes. A spike output is generated when the membrane potential surpasses a predefined threshold. The leaky-integrate-and-fire (LIF) model is commonly used to describe the dynamics of spiking neurons. In LIF, each neuron integrates the received spikes as the membrane potential $u_{\tau,i}$, which can be formulated as a first-order differential equation,

$$\textbf{LIF:} \ \bar{\lambda}\frac{du_\tau}{d\tau} = -(u_\tau - u_{rest}) + R \cdot I(\tau), \quad u_\tau < V_{th} , \tag{1}$$

where $u_\tau$ is the membrane potential, $I(\tau)$ is the input current, $V_{th}$ is the spiking threshold, and $R$ and $\bar{\lambda}$ are resistance and time constant, respectively. At time $\tau$, when the membrane potential $u_\tau$ reaches the spiking threshold $V_{th}$, a spike is triggered, and $u_\tau$ is reset to the resting potential $u_{\text{rest}}$, which is typically assigned a value of 0. We consider a simple current model $I_{\tau,i} = \sum_j w_{ij}s_{\tau,j} + b$, where $w_{ij}$ is the weight from neuron $j$ to neuron $i$. Then, the general form of LIF is described as:

$$u_{\tau+1,i} = \lambda(u_{\tau,i} - V_{th}s_{\tau,i}) + \sum_j w_{ij}s_{\tau,j} + b, \quad s_{\tau+1,i} = \mathbb{H}(u_{\tau+1,i} - V_{th}), \tag{2}$$

where $\mathbb{H}(x)$ is the Heaviside step function, which is the non-differentiable spiking function. $s_{\tau,i}$ is the binary spike train of neuron $i$, and $\lambda$ is a leaky term related to the constant $\bar{\lambda}$ in the LIF model. The constant $R$, $\bar{\lambda}$, and time step-size are absorbed into the weights $w_{ij}$ and bias $b$. The training of SNNs follows the process of BPTT, and the gradients are calculated with:

$$\frac{\partial \mathcal{L}}{\partial \boldsymbol{W}^l} = \sum_{\tau=1}^K \frac{\partial \mathcal{L}}{\partial \boldsymbol{s}_\tau^{l+1}} \frac{\partial \boldsymbol{s}_\tau^{l+1}}{\partial \boldsymbol{u}_\tau^{l+1}} \left( \frac{\partial \boldsymbol{u}_\tau^{l+1}}{\partial \boldsymbol{W}^l} + \sum_{k<\tau} \prod_{i=\tau-1}^k \left( \frac{\partial \boldsymbol{u}_{i+1}^{l+1}}{\partial \boldsymbol{u}_i^{l+1}} + \frac{\partial \boldsymbol{u}_{i+1}^{l+1}}{\partial \boldsymbol{s}_i^{l+1}} \frac{\partial \boldsymbol{s}_i^{l+1}}{\partial \boldsymbol{u}_i^{l+1}} \right) \frac{\partial \boldsymbol{u}_k^{l+1}}{\partial \boldsymbol{W}^l} \right), \tag{3}$$

where $\boldsymbol{W}^l$ is the trainable matrix on $l$-th layer and $\mathcal{L}$ is the loss. The terms $\frac{\partial \boldsymbol{s}_\tau^l}{\partial \boldsymbol{u}_\tau^l}$ are non-differentiable, and surrogate derivatives are typically used instead.

**Second-order SNNs**. The first-order neuron models assume that an input voltage spike causes an immediate change in synaptic current, affecting the membrane potential. However, in reality, a spike leads to the gradual release of neurotransmitters from the pre-synaptic neuron to the post-synaptic neuron. To capture this temporal dynamic, the synaptic conductance-based LIF model is used, which accounts for the gradual changes in input current over time. To solve this, Eshraghian et al. (2023) propose the second-order SNN, which is formulated as:

$$I_{\tau+1} = \alpha I_\tau + W X_{\tau+1}, \quad u_{\tau+1,i} = \beta u_{\tau,i} + I_{\tau+1,i} - R, \quad s_{\tau,i} = \mathbb{H}(u_{\tau+1,i} - V_{th}), \tag{4}$$

where $\alpha = exp(-\Delta t/\tau_{syn})$ and $\beta = exp(-\Delta t/\tau_{mem})$, $\tau_{syn}$ models the time constant of the synaptic current in an analogous way to how $\tau_{mem}$ models the time constant of the membrane potential.

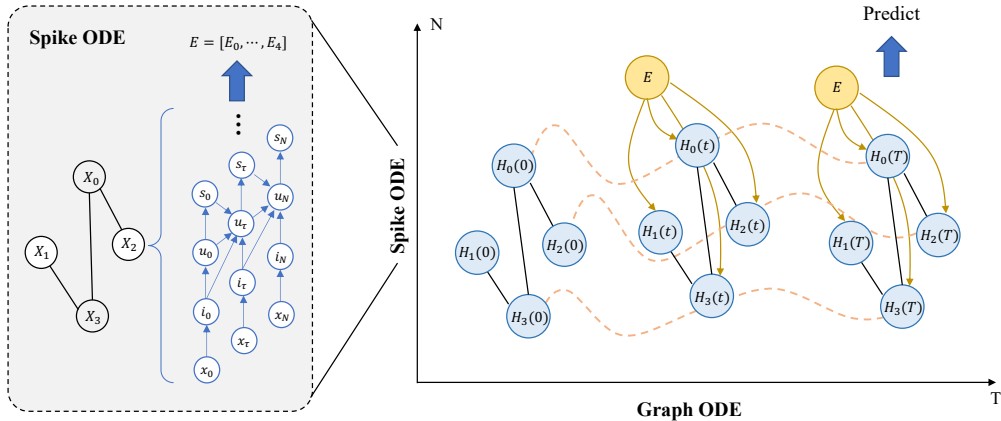

Figure 1: Overview of the proposed G-PDE. The model takes a graph with node features as input, which are initially encoded using the spike ODE (first-order or second-order). Subsequently, a differential graph equation (first-order or second-order) is employed to evolve the representation over time. Finally, the representation is projected for downstream tasks.

## 3.2 GRAPH ODE

Given a graph $G = (\mathcal{V}, \mathcal{E})$ with the node set $\mathcal{V}$ and the edge set $\mathcal{E}$. $\boldsymbol{X} \in \mathbb{R}^{|\mathcal{V}| \times d}$ is the node feature matrix, $d$ is the feature dimension. The binary adjacency matrix denoted as $\boldsymbol{A} \in \mathbb{R}^{|\mathcal{V}| \times |\mathcal{V}|}$, where $a_{ij} = 1$ denotes there exists a connection between node $i$ and $j$, and vice versa. Our goal is to learn a node representation $\boldsymbol{H}$ for downstairs tasks.

**First-order Graph ODE**. The first graph ODE is proposed by Xhonneux et al. (2020). Considering the Simple GCN Wu et al. (2019) with $\boldsymbol{H}_{n+1} = \boldsymbol{A}\boldsymbol{H}_n + \boldsymbol{H}_0$, the solution of the ODE is given by:

$$\frac{d\boldsymbol{H}(t)}{dt} = ln\boldsymbol{A}\boldsymbol{H}(t) + \boldsymbol{E}, \quad \boldsymbol{H}(t) = (\boldsymbol{A} - \boldsymbol{I})^{-1}(e^{(\boldsymbol{A}-\boldsymbol{I})t} - \boldsymbol{I})\boldsymbol{E} + e^{(\boldsymbol{A}-\boldsymbol{I})t}\boldsymbol{E}, \quad (5)$$

where $\boldsymbol{E} = \varepsilon(X)$ is the output of the encoder $\varepsilon$ and the initial value $\boldsymbol{H}(0) = (ln\boldsymbol{A})^{-1}(\boldsymbol{A} - \boldsymbol{I})\boldsymbol{E}$.

**Second-order Graph ODE**. The first second-order graph ODE is proposed by Rusch et al. (2022), which is represented as:

$$\boldsymbol{X}'' = \sigma(\boldsymbol{F}_\theta(\boldsymbol{X}, t)) - \gamma\boldsymbol{X} - \alpha\boldsymbol{X}', \quad (6)$$

where $(\boldsymbol{F}_\theta(\boldsymbol{X}, t))_i = \boldsymbol{F}_\theta(\boldsymbol{X}_i(t), \boldsymbol{X}_j(t), t)$ is a learnable coupling function with parameters $\theta$. Due to the unavailability of an analytical solution for Eq. 6, GraphCON Rusch et al. (2022) addresses it through an iterative numerical solver employing a suitable time discretization method. GraphCON utilizes the IMEX (implicit-explicit) time-stepping scheme, an extension of the symplectic Euler method Hairer et al. (1993) that accommodates systems with an additional damping term.

$$\boldsymbol{Y}^n = \boldsymbol{Y}^{n-1} + \Delta t[\sigma(\boldsymbol{F}_\theta(\boldsymbol{X}^{n-1}, t^{n-1})) - \gamma\boldsymbol{X}^{n-1} - \alpha\boldsymbol{Y}^{n-1}], \quad \boldsymbol{X}^n = \boldsymbol{X}^{n-1} + \Delta t\boldsymbol{Y}^n, \quad n = 1, \cdots, N,$$
$$(7)$$

where $\Delta t > 0$ is a fixed time-step and $\boldsymbol{Y}^n$, $\boldsymbol{X}^n$ denote the hidden node features at time $t^n = n\Delta t$.

## 4 METHOD

In this section, we present the proposed G-PDE for modeling the continuous graph ODE. It learns high-order spikes and continuous information to enhance the model capacity. Initially, we introduce a simplified version of G-PDE, namely the first-order graph PDE. Next, we derive the second-order spike representation and differentiation of the second-order SNN. Finally, we propose the second-order graph PDE network, which leverages a coupled high-order structure to extract additional information. More details can be found in Figure 1.

### 4.1 FIRST-ORDER GRAPH PDE NETWORK

The proposed G-PDE propagates along dual separate time dimensions, namely the spike ODE and graph ODE. We integrate these dual processes and reformulate them into a graph PDE form with corresponding parameters: the time step $t$ in the graph ODE and the time latency $\tau$ in the spike ODE. Intuitively, information is propagated interactively by both the SNN and graph ODE in dual time dimensions. We proceed to transform this process into a combination of two individual integral processes.

**Proposition 4.1** *Define the first-order SNNs ODE as $\frac{du_t^\tau}{d\tau} = g(u_t^\tau, \tau)$, and first-order Graph ODE as $\frac{du_t^\tau}{dt} = f(u_t^\tau, t)$, then the first-order graph PDE network can be formulated as:*

$$u_T^N = \int_0^T f\left(\int_0^N g(u_t^\tau)d\tau\right)dt = \int_0^N g\left(\int_0^T f(u_t^\tau)dt\right)d\tau, \quad \frac{\partial u_t^\tau}{\partial \tau} = g(u_t^\tau), \quad \frac{\partial u_t^\tau}{\partial t} = f(u_t^\tau).$$

where $u_t^\tau$ denotes the neuron membrane on latency $\tau$ and time step $t$, $N$ is the latency of SNNs, and $T$ is the time step length of graph ODE. Details of derivation are shown in the Appendix.

There are two perspectives to elucidate the proposition 4.1. Firstly, in terms of the temporal dimension ($0 \leq t \leq T$), the graph PDE initially computes the spike representation at each time step $t$, followed by evaluating the graph ODE process using this spike representation. Secondly, considering the latency dimension ($0 \leq \tau \leq N$), the graph PDE models the evolution of node embedding along a time series $t$, and subsequently integrates it across the latency dimension.

Furthermore, in our implementation of the first-order graph PDE, we adopt the first method by calculating the spike representation on each time step and then model the node embedding with Eqs. 5.

$$\frac{d\boldsymbol{H}(t)}{dt} = ln\boldsymbol{A}\boldsymbol{H}(t) + \frac{\sum_{\tau=1}^N \lambda^{N-\tau}s_\tau}{\sum_{\tau=1}^N \lambda^{N-\tau}}, \tag{8}$$

where $s_\tau$ is the binary spike on latency $\tau$, and $\lambda = exp(-\frac{\Delta t}{\kappa})$ with $\Delta t \ll \kappa$. $\boldsymbol{H}(t)$ is the spike representation on time step $t$, $\frac{\sum_{\tau=1}^N \lambda^{N-\tau}s_\tau}{\sum_{\tau=1}^N \lambda^{N-\tau}}$ is the initial encoding $\boldsymbol{E}$.

### 4.2 SECOND-ORDER SPIKING NEURAL NETWORKS

**Second-order SNNs forward propagation.** In our work, we set the forward propagation layer of SNNs to $L$. According to Eq. 4, the spiking dynamics for an SNN can be formulated as:

$$u^i(n) = \beta^i u^i(n-1) + (1-\beta^i)\frac{V_{th}^{i-1}}{\Delta t}\left(\alpha^i I^{i-1}(n-1) + \boldsymbol{W}^i s^{i-1}(n)\right) - V_{th}^i s^i(n), \tag{9}$$

where $i = 1, \cdots, L$ denotes the $i$-th layer, $s^0$ and $s^i$ denote the input and output of SNNs. $I^i$ is the input of the $i$-th layer, $n = 1, \cdots, N$ is the time step and $N$ is the latency. $\alpha^i = exp(-\Delta\tau/\tau_{syn}^i)$, $\beta^i = exp(-\Delta\tau/\tau_{mem}^i)$ and $0 < \Delta\tau \ll \{\tau_{syn}^i, \tau_{mem}^i\}$.

**Second-order spike representation.** Considering the second-order SNNs model defined by Eqs. 4, we define the weighted average input current as $\hat{I}(N) = \frac{1}{(\beta-\alpha)^2}\frac{\sum_{n=0}^{N-1}(\beta^{N-n}-\alpha^{N-n})I_{in}(n)}{\sum_{n=0}^{N-1}(\beta^{N-n}-\alpha^{N-n})}$, and the scaled weighted firing rate as $\hat{a}(N) = \frac{1}{\beta^2}\frac{V_{th}\sum_{n=0}^{N-1}\beta^{N-n}s(n)}{\sum_{n=0}^{N-1}(\beta^{N-n}-\alpha^{N-n})\Delta\tau}$. We use $\hat{a}(N)$ as the spiking representation of $\{s(n)\}_{n=1}^N$. Similarly to the first-order spiking representation Meng et al. (2022a), we directly determine the relationship between $\hat{I}(N)$ and $\hat{a}(N)$ using a differentiable mapping. Specifically, by combing Eq. 4, we have:

$$u(\tau+1) = \beta u(\tau) + \alpha I_{syn}(\tau) + I_{input}(\tau) - V_{th}s(\tau)$$

$$= \beta^2 u(\tau-k+1) + \alpha\sum_{i=0}^{k-1}\beta^i I_{syn}(\tau-i) + \sum_{i=0}^{k-1}\beta^i(I_{input}(\tau-i) - V_{th}s(\tau-i)). \tag{10}$$

By summing Eq. 10 over $n = 1$ to $N$, we have:

$$u(N) = \frac{1}{\beta - \alpha} \sum_{n=0}^{N-1} (\beta^{N-n} - \alpha^{N-n}) I_{in}(n) - \frac{1}{\beta} \sum_{n=0}^{N-1} \beta^{N-n} V_{th} s(n). \quad (11)$$

Dividing Eq. 11 by $\Delta\tau \sum_{n=0}^{N-1} (\beta^{N-n} - \alpha^{N-n})$, we can get:

$$\hat{a}(N) = \frac{\beta - \alpha}{\beta} \frac{\hat{I}(N)}{\Delta\tau} - \frac{u(N)}{\Delta\tau\beta \sum_{n=0}^{N-1}(\beta^{N-n} - \alpha^{N-n})} \approx \frac{\tau_{syn}\tau_{mem}}{\tau_{mem} - \tau_{syn}} \hat{I}(N) - \frac{u(N)}{\Delta\tau\beta \sum_{n=0}^{N-1}(\beta^{N-n} - \alpha^{N-n})}, \quad (12)$$

since $\lim\limits_{\Delta\tau \to 0} \frac{1 - \alpha/\beta}{\Delta\tau} = \frac{\tau_{syn}\tau_{mem}}{\tau_{mem} - \tau_{syn}}$ and $\Delta\tau \ll \frac{1}{\tau_{syn}} - \frac{1}{\tau_{mem}}$, we can approximate $\frac{\beta - \alpha}{\beta\Delta\tau}$ by $\frac{\tau_{syn}\tau_{mem}}{\tau_{mem} - \tau_{syn}}$. Following Meng et al. (2022a), and take $\hat{a}(N) \in [0, \frac{V_{th}}{\Delta\tau}]$ into consideration and assume $V_{th}$ is small, we ignore the term $\frac{u(N)}{\Delta\tau\beta \sum_{n=0}^{N-1}(\beta^{N-n} - \alpha^{N-n})}$, and approximate $\hat{a}(N)$ with:

$$\lim_{N \to \infty} \hat{a}(N) \approx clamp\left(\frac{\tau_{syn}\tau_{mem}}{\tau_{mem} - \tau_{syn}} \hat{I}(N), 0, \frac{V_{th}}{\Delta\tau}\right), \quad (13)$$

where $clamp(x, a, b) = max(a, min(x, b))$. During the training of the second-order SNNs, we have Proposition 4.2, which is similar to Meng et al. (2022a), and the detailed derivation is shown in the Appendix.

**Proposition 4.2** *Define* $\hat{a}^0(N) = \frac{\sum_{n=0}^{N-1} \beta_i^{N-n-2} s^0(n)}{\sum_{n=0}^{N-1} (\beta_i^{N-n} - \alpha_i^{N-n})\Delta\tau}$ *and* $\hat{a}^i(N) = \frac{V_{th}^i \sum_{n=0}^{N-1} \beta_i^{N-n-2} s^i(n)}{\sum_{n=0}^{N-1} (\beta_i^{N-n} - \alpha_i^{N-n})\Delta\tau}$, $\forall i = 1, \cdots, L$, *where* $\alpha^i = exp(-\Delta\tau/\tau_{syn}^i)$ *and* $\beta^i = exp(-\Delta\tau/\tau_{mem}^i)$. *Further, define the differentiable mappings*

$$\boldsymbol{z}^i = clamp\left(\frac{\tau_{syn}^i \tau_{mem}^i}{\tau_{mem}^i - \tau_{syn}^i} \boldsymbol{W}^i \boldsymbol{z}^{i-1}, 0, \frac{V_{th}^i}{\Delta\tau}\right), i = 1, \cdots, L. \quad (14)$$

*If* $\lim\limits_{N \to \infty} \hat{a}^i(N) = \boldsymbol{z}^i$ *for* $i = 0, 1, \cdots, L-1$, *then* $\hat{a}^{i+1}(N) \approx \boldsymbol{z}^{i+1}$ *when* $N \to \infty$.

**Differentiation on second-order spike representation**. With the forward propagation of the $i$-th layers, we get the output of SNN with $s^i = \{s^i(1), \cdots, s^i(N)\}$. We define the spike representation operator $r(s) = \frac{1}{\beta^2} \frac{V_{th} \sum_{n=0}^{N-1} \beta^{N-n} s(n)}{\sum_{n=0}^{N-1} (\beta^{N-n} - \alpha^{N-n})\Delta\tau}$, and get the final output $\boldsymbol{o}^L = r(s^L)$. For the simple second-order SNN, assuming the loss function as $\mathcal{L}$, we calculate the gradient $\frac{\partial \mathcal{L}}{\partial \boldsymbol{W}^i}$ as:

$$\frac{\partial \mathcal{L}}{\partial \boldsymbol{W}^i} = \frac{\partial \mathcal{L}}{\partial \boldsymbol{o}^i} \frac{\partial \boldsymbol{o}^i}{\partial \boldsymbol{W}^i} = \frac{\partial \mathcal{L}}{\partial \boldsymbol{o}^{i+1}} \frac{\partial \boldsymbol{o}^{i+1}}{\partial \boldsymbol{o}^i} \frac{\partial \boldsymbol{o}^i}{\partial \boldsymbol{W}^i}, \quad \boldsymbol{o}^i = r(s^i) \approx clamp\left(\boldsymbol{W}^i r(s^{i-1}), 0, \frac{V_{th}^i}{\Delta\tau}\right). \quad (15)$$

### 4.3 SECOND-ORDER GRAPH PDE NETWORK

Considering that higher-order spike ODEs preserve more information, higher-order graph ODEs effectively capture long-term dependencies in complex dynamic systems Luo et al. (2023). Therefore, to model graph evaluation, we leverage the efficient second-order graph PDEs while seamlessly integrating spike and graph ODEs. Although obtaining an analytical solution for the second-order PDE is not feasible, we can still derive a conclusion similar to Proposition 4.1.

**Proposition 4.3** *Define the second-order SNNs ODE as* $\frac{d^2 u_t^\tau}{d\tau^2} + \delta \frac{du_t^\tau}{d\tau} = g(u_t^\tau, \tau)$, *and second-order Graph ODE as* $\frac{d^2 u_t^\tau}{dt^2} + \gamma \frac{du_t^\tau}{dt} = f(u_t^\tau, t)$, *then the second-order graph PDE network is formulated as:*

$$u_t^\tau = \int_0^T h\left(\int_0^N e(u_t^\tau) d\tau\right) dt = \int_0^N e\left(\int_0^T h(u_t^\tau) dt\right) d\tau,$$

$$\frac{\partial^2 u_t^\tau}{\partial\tau^2} + \delta \frac{\partial u_t^\tau}{\partial\tau} = g(u_t^\tau), \quad \frac{\partial^2 u_t^\tau}{\partial t^2} + \gamma \frac{\partial u_t^\tau}{\partial t} = f(u_t^\tau),$$

*where* $e(u_t^\tau) = \int_0^N g(u_t^\tau) d\tau - \delta(u_t^N - u_t^0)$, $h(u_t^\tau) = \int_0^T f(u_t^\tau) dt - \gamma(u_T^\tau - u_0^\tau)$, $\frac{\partial e(u_t^\tau)}{\partial\tau} = g(u_t^\tau)$ *and* $\frac{\partial h(u_t^\tau)}{\partial t} = f(u_t^\tau)$.

Similarly to the first-order graph PDE, we implement the second-order graph PDE by calculating the spike representation on each time step and then model the node embedding with Eqs. 7.

**Differentiation on second-order graph PDE.** Denote the loss function as $\mathcal{L} = \sum_{i \in \mathcal{V}} \left| X_i^N - \bar{X}_i \right|^2$, and $\bar{X}_i$ is the label of node $i$. With the chain rule, we have: $\frac{\partial \mathcal{L}}{\partial W^l} = \frac{\partial \mathcal{L}}{\partial o_T^N} \frac{\partial o_T^N}{\partial o_T^l} \frac{\partial o_T^l}{\partial W^l}$. From Rusch et al. (2022), we have the conclusion that the traditional GNN model has the problem of gradient exponentially or vanishing, thus, we study the upper bound of the proposed G-PDE.

**Proposition 4.4** *Let $X^n$ and $Y^n$ be the node features, and $\Delta t \ll 1$. Then, the gradient of the graph ODE matrix $W_l$ is bounded as 16, and the gradient of the spike ODE $W^k$ is bounded as 16:*

$$\left| \frac{\partial \mathcal{L}}{\partial W_l} \right| \leq \frac{\beta' \hat{D} \Delta t (1 + \Gamma T \Delta t)}{v} \left( \max_{1 \leq i \leq v} (|X_i^0| + |Y_i^0|) \right) + \frac{\beta' \hat{D} \Delta t (1 + \Gamma T \Delta t)}{v} \left( \max_{1 \leq i \leq v} |\bar{X}_i| + \beta \sqrt{T \Delta t} \right)^2,$$

$$\left| \frac{\partial \mathcal{L}}{\partial W^k} \right| \leq \frac{(1 + T \Gamma \Delta t)(1 + N \Theta \Delta \tau) V_{th}}{v \beta^2 \Delta \tau} \left( \max_{1 \leq i \leq v} |X_i^N| + \max_{1 \leq i \leq v} |\bar{X}_i| \right). \qquad (16)$$

where $\beta = \max_x |\sigma(x)|$, $\beta' = \max_x |\sigma'(x)|$, $\hat{D} = \max_{i,j \in \mathcal{V}} \frac{1}{\sqrt{d_i d_j}}$, and $\Gamma := 6 + 4\beta' \hat{D} \max_{1 \leq n \leq T} ||W^n||_1$, $\Theta := 6 + 4\beta' \hat{D} \max_{1 \leq n \leq N} ||W^n||_1$. $d_i$ is the degree of node $i$, $\bar{X}_i$ is the label of node $i$. The first inequality can be obtained from Rusch et al. (2022) directly, and the derivation of the second inequality is presented in the Appendix.

The upper bound in Proposition 4.4 demonstrates that the total gradient remains globally bounded, regardless of the number of graph ODE layers $T$ and spike ODE layers $N$, as long as $\Delta t \sim T^{-1}$ and $\Delta \tau \sim N^{-1}$. This effectively addresses the issue of exploding gradients.

# 5 EXPERIMENTS

Our experimental evaluation comprehensively examines the effectiveness of the proposed framework in various graph learning tasks, including node classification and graph classification. We evaluate two settings of the method: G-PDE-1st-order, which utilizes the first-order SNN and second-order graph ODE, and G-PDE-2nd-order, which employs the second-order SNN and second-order graph ODE.

## 5.1 EXPERIMENTAL SETTINGS

**Datasets.** For the node classification, we evaluate G-PDE on homophilic (i.e., Cora McCallum et al. (2000), Citeseer Sen et al. (2008) and Pubmed Namata et al. (2012)) and heterophilic (i.e., Texas, Wisconsin and Cornell from the WebKB[1]) datasets, where high homophily indicates that a node's features are similar to those of its neighbors, and heterophily suggests the opposite. The homophily level is measured according to Pei et al. (2020), and is reported in Table 1 and 2. In the graph classification task, we utilize the MNIST dataset LeCun et al. (1998). To represent the grey-scale images as irregular graphs, we associate each superpixel (large blob of similar color) with a vertex, and the spatial adjacency between superpixels with edges. Each graph consists of a fixed number of 75 superpixels (vertices). To ensure consistent evaluation, we adopt the standard splitting of 55K-5K-10K for training, validation, and testing purposes.

**Baselines.** For the homophilic datasets, we use standard GNN baselines: GCN Kipf & Welling (2017), GAT Velickovic et al. (2017), MoNet Monti et al. (2017), GraphSage Hamilton et al. (2017), CGNN Xhonneux et al. (2020), GDE Poli et al. (2019), GRAND Chamberlain et al. (2021) and GraphCON Rusch et al. (2022). Due to the assumption on neighbor feature similarity does not hold in the heterophilic datasets, we utilize additional specific GNN methods as baselines: GPRGNN Chien et al. (2020), H2GCN Zhu et al. (2020a), GCNII Chen et al. (2020), Geom-GCN Pei et al. (2020) and PairNorm Zhao & Akoglu (2019). For the graph classification task, we apply ChebNet Defferrard et al. (2016), PNCNN Finzi et al. (2021), SplineCNN Fey et al. (2018), GIN Xu et al. (2019), and GatedGCN Bresson & Laurent (2017).

---

[1]http://www.cs.cmu.edu/afs/cs.cmu.edu/project/theo-11/www/wwkb/

**Implementation Details.** For the homophilic node classification task, we report the average results of 20 random initialization across 5 random splits. For the heterophilic node classification task, we present the average performance of the respective model over 10 fixed train/val/test splits. The results of baselines are reported in Rusch et al. (2022). In the implementation of G-PDE, we integrate the first and second-order SNN with the second-order graph ODE, referred to as G-PDE-1st-order and G-PDE-2nd-order, respectively. For G-PDE-1st-order, we set the hyperparameter $\lambda$ to 1. As for G-PDE-2nd-order, we assign the hyperparameters $\alpha$ and $\beta$ values of 1. During training, the hidden dimension is set to 64, the dropout ratio set to 0.3, and we use the Adam Kingma & Ba (2014) optimizer and set the batch size to 1024. The number of training epochs is 200, and the learning rate is set to 0.001.

Table 1: The test accuracy (in %) for node classification on homophilic datasets. The results are calculated by averaging the results of 20 random initializations across 5 random splits. The mean and standard deviation of these results are obtained. **Bold** numbers means the best performance, and underline numbers indicates the second best performance.

| Homophily level | Cora 0.81 | Citeseer 0.74 | Pubmed 0.80 |
|---|---|---|---|
| GAT-ppr | 81.6±0.3 | 68.5±0.2 | 76.7±0.3 |
| MoNet | 81.3±1.3 | 71.2±2.0 | 78.6±2.3 |
| GraphSage-mean | 79.2±7.7 | 71.6±1.9 | 77.4±2.2 |
| GraphSage-maxpool | 76.6±1.9 | 67.5±2.3 | 76.1±2.3 |
| CGNN | 81.4±1.6 | 66.9±1.8 | 66.6±4.4 |
| GDE | 78.7±2.2 | 71.8±1.1 | 73.9±3.7 |
| GCN | 81.5±1.3 | 71.9±1.9 | 77.8±2.9 |
| GAT | 81.8±1.3 | 71.4±1.9 | 78.7±2.3 |
| GRAND | 83.6±1.0 | 73.4±0.5 | 78.8±1.7 |
| GraphCON-GCN | 81.9±1.7 | 72.9±2.1 | 78.8±2.6 |
| GraphCON-GAT | 83.2±1.4 | 73.2±1.8 | 79.5±1.8 |
| G-PDE-1st-order | 83.3±2.1 | 73.7±2.0 | 76.9±2.7 |
| G-PDE-2nd-order | **83.7±1.3** | **75.2±2.0** | **79.6±2.3** |

## 5.2 PERFORMANCE COMPARISION

**Homophilic node classification**. Table 1 shows the results of the proposed G-PDE with the comparison of the baselines. From the results, we find that: (1) Compared with the discrete methods, the continuous methods achieve better performance, indicating that the continuous methods would help to capture the continuous changes and subtle dynamics from graphs. (2) The first and second-order G-PDE outperforms other baselines in most cases, which shows G-PDE preserves the advantages of spike and graph ODE for graph learning. (3) The second-order G-PDE consistently outperforms the first-order, we attribute the reason that high-order structure would preserve more information than the first-order G-PDE.

Table 2: The test accuracy (in %) for node classification on heterophilic datasets. All results represent the average performance of the respective model over 10 fixed train/val/test splits. **Bold** numbers means the best performance, and underline numbers indicates the second best performance.

| Homophily level | Texas 0.11 | Wisconsin 0.21 | Cornell 0.30 |
|---|---|---|---|
| GPRGNN | 78.4±4.4 | 82.9±4.2 | 80.3±8.1 |
| H2GCN | 84.9±7.2 | 87.7±5.0 | 82.7±5.3 |
| GCNII | 77.6±3.8 | 80.4±3.4 | 77.9±3.8 |
| Geom-GCN | 66.8±2.7 | 64.5±3.7 | 60.5±3.7 |
| PairNorm | 60.3±4.3 | 48.4±6.1 | 58.9±3.2 |
| GraphSAGE | 82.4±6.1 | 81.2±5.6 | 76.0±5.0 |
| MLP | 80.8±4.8 | 85.3±3.3 | 81.9±6.4 |
| GCN | 55.1±5.2 | 51.8±3.1 | 60.5±5.3 |
| GAT | 52.2±6.6 | 49.4±4.1 | 61.9±5.1 |
| GraphCON-GCN | 85.4±4.2 | 87.8±3.3 | **84.3±4.8** |
| GraphCON-GAT | 82.2±4.7 | 85.7±3.6 | 83.2±7.0 |
| G-PDE-1st-order | 81.6±6.2 | 84.9±3.2 | 80.4±1.9 |
| G-PDE-2nd-order | **87.3±4.2** | **88.8±2.5** | 83.7±2.7 |

**Heterophilic node classification**. We show the results of different methods in Table 2, and we have the following observation: (1) The traditional massage-passing-based methods perform worse than the well-designed methods for heterophilic datasets, especially for GCN and GAT, this is because the assumption that the neighbor feature is similar does not hold. (2) The G-PDE-2nd-order still outperforms G-PDE-1st-order, which proves the effectiveness of high-order structure again. (3) The G-PDE-1st-order performs worse than GraphCON, we attribute the reason that the first-order G-PDE loses the features of the heterophilic dataset for node discriminate, which is more important than the homophilic.

Table 4: Ablation results. **Bold** numbers means the best performance, and underline numbers indicates the second best performance.

| *Homophily level* | Cora 0.81 | Citeseer 0.74 | Pubmed 0.80 | Texax 0.11 | Wisconsin 0.21 | Cornell 0.3 |
|---|---|---|---|---|---|---|
| G-PDE-1st-1st | 82.8±0.8 | 72.5±1.3 | 75.4±1.7 | 81.1±4.4 | 83.8±3.1 | 80.7±2.6 |
| G-PDE-2nd-1st | 83.5±1.8 | 73.4±2.1 | 77.2±2.3 | 83.1±3.8 | 84.4±2.2 | 81.2±2.7 |
| G-PDE-1st-order | 83.3±2.1 | 73.7±2.0 | 76.9±2.7 | 81.6±6.2 | 84.9±3.2 | 80.4±1.9 |
| G-PDE-2nd-order | **83.7±1.3** | **75.2±2.0** | **79.6±2.3** | **87.3±4.2** | **88.8±2.5** | **83.7±2.7** |

**Graph classification**. Table 3 shows the comparison of the graph classification results. From the results, we find that: (1) The first- and second-order graph PDE outperforms the traditional GNNs; we stress that the continuous process both in the SNN and graph ODE would help to extract the continuous changes and subtle dynamics from graphs. (2) The first-order G-PDE performs worse than the second-order G-PDE, indicating that the high-order structure would help to preserve more information than the first-order, and will not take much additional overhead. (3) The first-order method performs worse than GraphCON-GAT and better than GraphCON-GCN. We attribute this

Table 3: The test accuracy (in %) for graph classification on MNIST datasets. **Bold** numbers means the best performance, and underline numbers indicates the second best performance.

| Model | Test accuracy |
|---|---|
| ChebNet Defferrard et al. (2016) | 75.62 |
| MoNet Monti et al. (2017) | 91.11 |
| PNCNN Finzi et al. (2021) | 98.76 |
| SplineCNN Fey et al. (2018) | 95.22 |
| GIN Xu et al. (2019) | 97.23 |
| GatedGCN Bresson & Laurent (2017) | 97.95 |
| GCN Kipf & Welling (2017) | 88.89 |
| GAT Velickovic et al. (2017) | 96.19 |
| GraphCON-GCN Rusch et al. (2022) | 98.68 |
| GraphCON-GAT Rusch et al. (2022) | 98.91 |
| G-PDE-1st-order | 98.82 |
| G-PDE-2nd-order | **98.92** |

difference to the fact that although the spike ODE loses some information for graph representation, this part of the information is not crucial, and we can compensate for it through other technical means, such as GAT.

## 5.3 ABLATION STUDY

We conducted ablation studies to assess the contributions of different components using two variants, and the results are presented in Table 4. Specifically, we introduced two model variants: (1) G-PDE-1st-1st, which utilizes the first-order spike ODE and first-order graph ODE, and (2) G-PDE-2nd-1st, incorporating the second-order spike ODE and first-order graph ODE. Table 4 reveals that (1) G-PDE-2nd-order consistently outperforms other variations, while G-PDE-1st-1st yields the worst performance, validating the effectiveness of our proposed method. (2) In most cases, G-PDE-2nd-1st exhibits better performance than G-PDE-1st-order, suggesting that the second-order spike ODE contributes more to representation than the second-order graph ODE.

## 6 CONCLUSION

In this paper, we address the practical problem of continuous graph representation learning and propose an effective method named G-PDE. G-PDE incorporates the spike ODE and graph ODE into a unified graph PDE from two distinct time dimensions, which preserves the advantages of low-power consumption and fine-grained feature extraction. Considering that the high-order structure would help to relieve the problem of information loss, we propose the second-order G-PDE and derive the second-order spike representation. To ensure the stability of G-PDE, we further prove that G-PDE mitigates the gradient exploding and vanishing problem. Extensive experiments on diverse datasets validate the efficacy of proposed G-PDE compared with various competing methods. In future works, we will extend G-PDE to higher-order structure to explore more efficient structure for continuous graph learning.

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

## A  PROOF OF PROPOSITION 4.1

**Proposition 4.1** *Define the first-order SNNs ODE as $\frac{du_t^\tau}{d\tau} = g(u_t^\tau, \tau)$, and first-order Graph ODE as $\frac{du_t^\tau}{dt} = f(u_t^\tau, t)$, then the first-order graph PDE network can be formulated as:*

$$u_t^\tau = \int_0^T f\left(\int_0^N g(u_t^\tau)d\tau\right)dt = \int_0^N g\left(\int_0^T f(u_t^\tau)dt\right)d\tau.$$

*Proof.*

$$\frac{du_t^\tau}{d\tau} = g(u_t^\tau, \tau), \quad \frac{du_t^\tau}{dt} = f(u_t^\tau, t),$$

$u_t^\tau$ is a function related to variable $t$ and $\tau$, we have $\frac{\partial u_t^\tau}{\partial \tau} = g(u_t^\tau)$ and $\frac{\partial u_t^\tau}{\partial t} = f(u_t^\tau)$. Thus,

$$u_t^\tau = \int_0^T f\left(\int_0^N g(u_t^\tau)d\tau\right)dt = \int_0^N g\left(\int_0^T f(u_t^\tau)dt\right)d\tau.$$

## B  PROOF OF EQ. 13

*Proof.* From Eq. 10, we have:

$$u(\tau + 1) = \beta^2 u(\tau - k + 1) + \alpha \sum_{i=0}^{k-1} \beta^i I_{syn}(\tau - i) + \sum_{i=0}^{k-1} \beta^i (I_{input}(\tau - i) - V_{th}s(\tau - i)), \quad (17)$$

$$u(N) = \alpha \sum_{n=0}^{N-1} \beta^n I_{syn}(N - n - 1) + \sum_{n=0}^{N-1} \beta^n (I_{input}(N - n - 1) - V_{th}s(N - n - 1)). \quad (18)$$

Due to:

$$I_{syn}(\tau + 1) = \alpha^k I_{syn}(\tau - k + 1) + \sum_{i=0}^{k} \alpha^i I_{input}(\tau - i), \quad (19)$$

we have,

$$
\begin{aligned}
u(N) =& \alpha \sum_{n=0}^{N-1} \beta^{N-n-1} I_{syn}(n) + \sum_{n=0}^{N-1} \beta^{N-n-1}\left(I_{input}(n) - V_{th}s(n)\right) \\
=& \alpha \left(\left(\frac{\beta^{N-1}\alpha^{-1}\left(1 - (\frac{\alpha}{\beta})^N\right)}{1 - \frac{\alpha}{\beta}}\right) I_{in}(0) + \left(\frac{\beta^{N-2}\alpha^{-1}\left(1 - (\frac{\alpha}{\beta})^{N-1}\right)}{1 - \frac{\alpha}{\beta}}\right) I_{in}(1) + \cdots \right. \\
& \left. + \left(\frac{\beta^{N-i}\alpha^{-1}\left(1 - (\frac{\alpha}{\beta})^{N-i+1}\right)}{1 - \frac{\alpha}{\beta}}\right) I_{in}(i-1) + \cdots + (\beta^2\alpha^{-1} + \beta + \alpha)I_{in}(N-3) \right. \\
& \left. + (\beta\alpha^{-1} + 1)I_{in}(N-2) + \alpha^{-1}I_{in}(N-1)\right) - \sum_{n=0}^{N-1} \beta^{N-n-1}V_{th}s(n) \\
=& \frac{1}{\beta - \alpha}\left(\left(\beta^N\left(1 - \left(\frac{\alpha}{\beta}\right)^N\right)I_{in}(0)\right) + \cdots + \left(\beta^{N-i+1}\left(1 - \left(\frac{\alpha}{\beta}\right)^{N-i+1}\right)I_{in}(i-1)\right)\right. \\
& \left. + \cdots + (\beta - \alpha)I_{in}(N-1)\right) - \sum_{n=0}^{N-1} \beta^{N-n-1}V_{th}s(n) \\
=& \frac{1}{\beta - \alpha}\sum_{n=0}^{N-1}(\beta^{N-n} - \alpha^{N-n})I_{in}(n) - \sum_{n=0}^{N-1} \beta^{N-n-1}V_{th}s(n).
\end{aligned}
$$

Define $\hat{I}(N) = \frac{1}{(\beta-\alpha)^2} \frac{\sum_{n=0}^{N-1}(\beta^{N-n}-\alpha^{N-n})I_{in}(n)}{\sum_{n=0}^{N-1}(\beta^{N-n}-\alpha^{N-n})}$, and $\hat{a}(N) = \frac{1}{\beta^2} \frac{V_{th}\sum_{n=0}^{N-1}\beta^{N-n}s(n)}{\sum_{n=0}^{N-1}(\beta^{N-n}-\alpha^{N-n})}$, we have:

$$\hat{a}(N) = \frac{\beta-\alpha}{\beta}\frac{\hat{I}(N)}{\Delta\tau} - \frac{u(N)}{\Delta\tau\beta\sum_{n=0}^{N-1}(\beta^{N-n}-\alpha^{N-n})} \approx \frac{\tau_{syn}\tau_{mem}}{\tau_{mem}-\tau_{syn}}\hat{I}(N) - \frac{u(N)}{\Delta\tau\beta\sum_{n=0}^{N-1}(\beta^{N-n}-\alpha^{N-n})},$$

where $\alpha = exp(-\Delta\tau/\tau_{syn})$, $\beta = exp(-\Delta\tau/\tau_{mem})$.

## C  PROOF OF PROPOSITION 4.3

**Proposition 4.3** *Define the second-order SNNs ODE as $\frac{d^2u_t^\tau}{d\tau^2} + \delta\frac{du_t^\tau}{d\tau} = g(u_t^\tau, \tau)$, and second-order Graph ODE as $\frac{d^2u_t^\tau}{dt^2} + \gamma\frac{du_t^\tau}{dt} = f(u_t^\tau, t)$, then the second-order graph PDE network is formulated as:*

$$u_t^\tau = \int_0^T h\left(\int_0^N e(u_t^\tau)d\tau\right)dt = \int_0^N e\left(\int_0^T h(u_t^\tau)dt\right)d\tau,$$

$$\frac{\partial^2 u_t^\tau}{\partial\tau^2} + \delta\frac{\partial u_t^\tau}{\partial\tau} = g(u_t^\tau), \quad \frac{\partial^2 u_t^\tau}{\partial t^2} + \gamma\frac{\partial u_t^\tau}{\partial t} = f(u_t^\tau),$$

*where $e(u_t^\tau) = \int_0^N g(u_t^\tau)d\tau - \delta(u_t^N - u_t^0)$, and $h(u_t^\tau) = \int_0^T f(u_t^\tau)dt - \gamma(u_T^\tau - u_0^\tau)$.*

*Proof.* Obviously,

$$\frac{\partial^2 u_t^\tau}{\partial\tau^2} + \delta\frac{\partial u_t^\tau}{\partial\tau} = g(u_t^\tau), \quad \frac{\partial^2 u_t^\tau}{\partial t^2} + \gamma\frac{\partial u_t^\tau}{\partial t} = f(u_t^\tau),$$

so, $\quad \frac{\partial u_t^\tau}{\partial\tau} + \delta(u_t^N - u_t^0) = \int_0^N g(u_t^\tau)d\tau, \quad \frac{\partial u_t^\tau}{\partial t} + \gamma(u_T^\tau - u_0^\tau) = \int_0^T f(u_t^\tau)dt.$

Define $e(u_t^\tau) = \int_0^N g(u_t^\tau)d\tau - \delta(u_t^N - u_t^0)$, and $h(u_t^\tau) = \int_0^T f(u_t^\tau)dt - \gamma(u_T^\tau - u_0^\tau)$, we have:

$$\frac{\partial u_t^\tau}{\partial\tau} = e(u_t^\tau), \quad \frac{\partial u_t^\tau}{\partial t} = h(u_t^\tau),$$

thus,

$$u_t^\tau = \int_0^T h\left(\int_0^N e(u_t^\tau)d\tau\right)dt = \int_0^N e\left(\int_0^T h(u_t^\tau)dt\right)d\tau,$$

where $\frac{\partial e(u_t^\tau)}{\partial\tau} = g(u_t^\tau)$ and $\frac{\partial h(u_t^\tau)}{\partial t} = f(u_t^\tau)$.

## D  PROOF OF PROPOSITION 4.4

$$\frac{\partial\mathcal{L}}{\partial\boldsymbol{W}^k} = \frac{\partial\mathcal{L}}{\partial\boldsymbol{Z}_T^N}\frac{\partial\boldsymbol{Z}_T^N}{\partial\boldsymbol{Z}_l^N}\frac{\partial\boldsymbol{Z}_l^N}{\partial\boldsymbol{W}^k} = \frac{\partial\mathcal{L}}{\partial\boldsymbol{Z}_T^N}\prod_{n=l+1}^T\frac{\partial\boldsymbol{Z}_n^N}{\partial\boldsymbol{Z}_{n-1}^N}\frac{\partial\boldsymbol{Z}_l^N}{\partial\boldsymbol{W}^k}$$

$$= \frac{\partial\mathcal{L}}{\partial\boldsymbol{Z}_T^N}\prod_{n=l+1}^T\frac{\partial\boldsymbol{Z}_n^N}{\partial\boldsymbol{Z}_{n-1}^N}\frac{\partial\boldsymbol{Z}_l^N}{\partial\boldsymbol{Z}_l^k}\frac{\partial\boldsymbol{Z}_l^k}{\partial\boldsymbol{W}^k}$$

$$= \frac{\partial\mathcal{L}}{\partial\boldsymbol{Z}_T^N}\prod_{n=l+1}^T\frac{\partial\boldsymbol{Z}_n^N}{\partial\boldsymbol{Z}_{n-1}^N}\prod_{i=k+1}^N\frac{\partial\boldsymbol{Z}_l^i}{\partial\boldsymbol{Z}_l^{i-1}}\frac{\partial\boldsymbol{Z}_l^k}{\partial\boldsymbol{W}^k},$$

From Rusch et al. (2022), we have:

$$\left\|\frac{\partial\mathcal{L}}{\partial\boldsymbol{Z}_T^N}\right\|_\infty \leq \frac{1}{v}\left(\max_{1\leq i\leq v}|\boldsymbol{X}_i^N| + \max_{1\leq i\leq v}|\bar{\boldsymbol{X}}_i|\right), \quad \left\|\frac{\partial\boldsymbol{Z}_T^N}{\partial\boldsymbol{Z}_t^N}\right\|_\infty \leq 1 + T\Gamma\Delta t. \tag{20}$$

Due to the second-order SNN has a similar formulation to second-order GNN, we have a similar conclusion,

$$\left\|\frac{\partial\boldsymbol{Z}_l^N}{\partial\boldsymbol{Z}_l^k}\right\|_\infty \leq 1 + N\Theta\Delta\tau, \tag{21}$$

with $\beta = \max_{x} |\sigma(x)|$, $\beta^{'} = \max_{x} |\sigma^{'}(x)|$, $\hat{D} = \max_{i,j \in \mathcal{V}} \frac{1}{\sqrt{d_i d_j}}$, and $\Theta := 6 + 4\beta^{'} \hat{D} \max_{1 \leq n \leq N} ||\boldsymbol{W}^n||_1.$, and with Eq. 15, we have:

$$\frac{\partial \boldsymbol{Z}_l^k}{\partial \boldsymbol{W}^k} \approx r(\boldsymbol{Z}_l^{k-1}) \leq \frac{V_{th}}{\beta^2 \Delta \tau}. \tag{22}$$

Multipling 20, 21 and 22, we have the upper bound:

$$\frac{\partial \mathcal{L}}{\partial \boldsymbol{W}^k} \leq \frac{(1 + T\Gamma\Delta t)(1 + N\Theta\Delta\tau)V_{th}}{v\beta^2\Delta\tau} \left( \max_{1 \leq i \leq v} |\boldsymbol{X}_i^N| + \max_{1 \leq i \leq v} |\bar{\boldsymbol{X}}_i| \right). \tag{23}$$

