# OpenReview forum: "Graph-PDE: Coupled ODE Structure for Graph Neural Networks"
_ICLR.cc/2024/Conference — ICLR 2024 Conference Withdrawn Submission_

### Official Review · Reviewer_9J7f · 2023-10-24

**Soundness:** 3 good
**Presentation:** 3 good
**Contribution:** 4 excellent
**Rating:** 8
**Confidence:** 4

**Summary:**

The paper introduces a novel architecture called the Graph-PDE approach that combines spike and graph ordinary differential equations within a unified graph neural network partial differential equation framework. The approach addresses the challenges of incorporating spike neural networks and graph ODEs into a unified model, ensuring stability and solving the exploding and vanishing gradient problem. The authors propose a second-order spike representation and introduce the second-order G-PDE to improve low-power consumption in SNNs. The stability of G-PDE is theoretically analyzed, and the framework is evaluated on various graph-based learning tasks, outperforming state-of-the-art methods.

**Strengths:**

1.This paper incorporates the coupled ODE (spike ODE and graph ODE) into a unified graph PDE, which reserves the advantages of low-power consumption from spike ODE and extraction of continuous changes and subtle dynamics from graph ODE. In addition, the propose the second-order graph PDE, which would alleviate the information loss issue in the first-order PDE.
2. This paper first derive the second-order spike representation and study the backpropagation of second-order spike ODE, which is further applied in the second-order graph PDE.
3.This paper the stability of the proposed G-PDE, and prove that G-PDE mitigates the problem of exploding and vanishing gradients and improves the trainingability of deep multilayer GNNs.
4. The paper is well-written and effectively explains the concepts, methodologies, and experimental results. The organization of the document allows for easy understanding of the proposed approach.

**Weaknesses:**

1.Formatting problem, Eqs 12, 16 exceeds the width of the paper.
2.Is there any connection between time series t in graph ODE and time latency \tau in SNN ODE? In other word, is it necessary to consider the circumstance of ?
3.Lack the reference of graph ODE, such as [1,2,3,4]

Reference:
[1] CF-GODE: Continuous-Time Causal Inference for Multi-Agent Dynamical Systems
[2] Generalizing Graph ODE for Learning Complex System Dynamics across Environments
[3] Neural temporal walks: Motif-aware representation learning on continuous-time dynamic graphs
[4] Continuous Temporal Graph Networks for Event-Based Graph Data

**Questions:**

1. Can the second-order spike representation be applied to other types of neural networks beyond SNNs? If so, are there any potential benefits in terms of model performance?

---

### Official Review · Reviewer_d8xw · 2023-10-31

**Soundness:** 2 fair
**Presentation:** 3 good
**Contribution:** 3 good
**Rating:** 6
**Confidence:** 2

**Summary:**

In this paper, the authors tackle the challenge of continuous graph representation learning by combining two distinct approaches: the spike ODE and the graph ODE. Both of these models come with their own limitations. The spike ODE, for instance, is inherently discrete and struggles to handle continuous changes effectively. On the other hand, the graph ODE encounters the notorious issue of exploding and vanishing gradients.

To overcome these drawbacks, the authors introduce a novel model that unifies these two methods into a coherent graph PDE framework. This novel approach leverages the low computational cost of the Spike ODE while also enabling the extraction of continuous variations from the Graph ODE. Additionally, the authors introduce second-order spike representations and integrate them into a second-order graph PDE. Notably, their model successfully mitigates the gradient-related challenges, demonstrating its stability.

To substantiate the effectiveness of their proposed model, the authors conduct thorough evaluations alongside multiple other existing models. This comprehensive analysis serves to validate the enhanced performance and capabilities of their novel approach.

**Strengths:**

Thank you for your interest in ICLR! This paper explores a relevant topic in the context of the ICLR Conference. While both Spike ODE and graph PDE have their own inherent drawbacks, the authors take an innovative approach by merging these two techniques to create a new model that attempts to combine their strengths.

The paper does provide mathematical proofs to support the proposed model, which is a crucial aspect of establishing its validity. Moreover, the authors undertake an extensive evaluation process, comparing their graph PDE model against various existing models. This empirical analysis offers insights into the model's performance and its potential practical applications, though it is worth noting that further research may be needed to address certain limitations or challenges.

**Weaknesses:**

One area where the paper could be improved is in the level of detail provided regarding the design of the graph-PDE model. To enhance the clarity and transparency of the research, it would be valuable for the authors to offer a more comprehensive description of the model's structure, particularly in the context of the evaluations. This additional information could further aid readers in grasping the methodology and potentially facilitate better reproducibility and assessment by the research community.

**Questions:**

I have a couple of suggestions for the authors that may enhance the paper. First, it would be beneficial to include visual aids, such as charts or graphs, to illustrate the results. Visual representations can make the findings more accessible and easier to interpret for readers.

Secondly, conducting an evaluation specifically focused on gradients could be a valuable addition. This could provide insights into the behavior of gradients in the proposed model and help further validate its stability and performance.

---

### Official Review · Reviewer_KQxP · 2023-10-31

**Soundness:** 3 good
**Presentation:** 2 fair
**Contribution:** 3 good
**Rating:** 3
**Confidence:** 5

**Summary:**

This paper proposes a coupled ODE framework that combines both SNN and GraphODE to jointly model the dynamics in the latency dimension and time dimension. To capture the long-term dependency and preserve more information in existing SNN models, the authors propose a second-order spike representation and derive its backpropagation. They further theoretically show the stability of the proposed G-PDE which is able to alleviate the gradient exploding and vanishing problems. Finally, they conduct experiments on static graph learning tasks such as node classification, showing G-PDE is able to surpass compared baselines.

**Strengths:**

The paper introduces a framework that combines SNN with GraphODE. The major novelty lies in the derivation of second-order representation of SNN and theoretical analysis of the stability G-PDE. The experiments compared G-PDE with existing baselines over the static graph learning task, e.g. node representation, showing its superior performance.

**Weaknesses:**

1. My major concern is the motivation for combining SNN with GraphODE.  The advantage of SNN is the low power consumption while GraphODE is good for capturing continuous dynamics over time. However, the designed model G-ODE is evaluated only on static graph tasks, e.g. node classification, instead of on modeling the dynamic evolution of graphs. Then why it is important to use GraphODE on such tasks? If it is just trying to alleviate the over-smoothing issue as studied in continuous graph neural network [1], it was not listed as a baseline for comparison and the storyline is not compatible with the experiment results (the storyline is about a dynamical system, experiment results are static graphs).

2. Second, the authors developed second-order representations of SNN, motivated by the higher-order ODE [2] to capture long-term dependencies in complex dynamic systems (page 6 ). Again, the experiment is not targeted at dynamic graphs so why there's a need to develop higher-order SNN? If it is to capture multi-hop neighbors of a node, wouldn't that sacrifice its original advantage of being low-power consumption?

3. The citation format seems to be wrong. There should be brackets around.




[1] Louis-Pascal Xhonneux, Meng Qu, Jian Tang. Continuous graph neural networks.
[2] Xiao Luo, et.al. HOPE: High-order Graph ODE For Modeling Interacting Dynamics.

**Questions:**

1. Can the authors provide motivation examples to illustrate why they need to combine SNN with GraphODE on static graph learning task?
2. The overall writing can be more organized, including citation format.

---

### Official Review · Reviewer_w3bP · 2023-11-01

**Soundness:** 2 fair
**Presentation:** 1 poor
**Contribution:** 2 fair
**Rating:** 3
**Confidence:** 4

**Summary:**

This paper investigates the problem of continuous graph learning and propose G-PDE, which incorporates Spike Neural Networks (SNN) and graph ODE into a unified graph PDE. G-PDE leverages the advantages of high-order spike representations, effectively addressing challenges such as information loss and model stability during training. Extensive experiments conducted on various datasets show the efficacy of the proposed framework.

**Strengths:**

1.	The ides of incorporating spike ODE (discrete solution) with graph ODE (continuous framework) as a unified PDE framework is novel to me.
2.	Extensive and thorough experiments have been conducted for model evaluation.

**Weaknesses:**

1.	The motivation provided in the paper is somewhat unclear. Essentially, GraphCON[1] is described as a "physically inspired" approach that formulates the stacking of GNN layers as discretized ODE governed by graph-coupled oscillators. In G-PDE, SNN act as the role of encoder within GraphCON[1] framework. However, in introduction section, we can only tell that the drawbacks of SNN arises from dealing with time-series problem. The justification why SNN should replace general GNN encoder, and their connection are both lacking.
2.	The author claims that G-PDE is intentionally designed to leverage the advantages of both spike ODE and graph ODE. However, it should be noted that the fundamental structure of G-PDE is an incremental improvement upon GraphCON[1], where the static encoder is substituted by SNN.
3.	The paper is not easy to follow due to several reasons: (1) notation abuse; (2) lack of model details. For details and relevant questions, please refer to the Questions sections.
4.	Based on the experimental evaluations, the improvements brought by G-PDE-2nd-order appear to be minimal across datasets such as Cora, PubMed and MNIST. Furthermore, G-PDE-1st-order even underperforms variants of GraphCON[1] in almost all scenarios. Given that G-PDE can be seen as a variant of GraphCON[1] with a sophisticated SNN encoder, there are doubts regarding the effectiveness of introducing SNNs.


[1] Rusch, T. Konstantin, et al. "Graph-coupled oscillator networks." ICML 2022.

**Questions:**

I have some concerns regarding the presentations of this papers:

1.	Notations abuses
    - a.	What’s the definition of Greek letter ‘k’ right after Eq. (8)
    - b.	What’s the definition of ‘k’ in Eq. (10)
    - c.	The definition of r(s) in Eq. (15) is identical to \hat{\alpha}(N) before Eq. (10), why should we define two identical variables?
2.	Model details
    - a.	Section 4.1 only introduces the first-order G-PDE constructed by first-order SNN and first-order graph ODE. However, the variant G-PDE-1st-order in the experimental section is a combination of first-order SNN and second-order graph ODE, which is missing in the main context. Where’s the corresponding model details for this variant?
    - b.	Prop. 4.1 appears to be a straightforward formulation based on simple double integration, rather than a substantial proposition. In addition, to be more rigorous, the prerequisite of double integration exchangeability can be also included in the claim.
    - c.	To enhance readability, it would be helpful to provide the detailed derivation from Eq. (4) to Eq. (9)? In addition, how does the author derive the second line from the first line in Eq. (10)?
    - d.	In G-PDE-2nd-order, does SNN replace the F_{\theta} in Eq. (7)? If so, its connection towards model inputs (X^{n -1}) is entirely missing in your derivation from Eq. (9) to (15). Can you clarify how SNN calculate the spike representations using X^{n -1}?
    - e.	Prop. 4.3 is not convincing. Per your proof in appendix, the left-hand side of equation in second line (the integration of second-order graph ODE) is incorrect due to the absence of lower & upper bound of first-order derivative (first term). Consequently, the equations in third line (e(\cdot) & h(\cdot)) are also incorrect, which makes the second-order G-PDE framework unconvincing.

**Details Of Ethics Concerns:**

None.

---

### Official Review · Reviewer_mPJA · 2023-11-04

**Soundness:** 3 good
**Presentation:** 3 good
**Contribution:** 3 good
**Rating:** 6
**Confidence:** 4

**Summary:**

This article proposes a new Graph-PDE framework to solve the graph classification problem in low energy consumption scenarios. Graph-PDE combines spike and graph ODE in a unified graph partial differential equation, and proposed the first-order and second-order Graph-PDE. Experiment results show the competitive performance of G-PDE compared to state-of-the-art methods.

**Strengths:**

1.	Novel Architecture. This paper incorporates the coupled ODE (spike ODE and graph ODE) into a unified graph PDE, which reserves the advantages of low-power consumption from spike ODE and extraction of continuous changes and subtle dynamics from graph ODE. In addition, this paper propose the second-order graph PDE, which would alleviate the information loss issue in the first-order PDE.
2.	Second-order Spike Representation. This paper first derive the second-order spike representation and study the backpropagation of second-order spike ODE, which is further applied in the second-order graph PDE.
3.	Theoretical Analysis of the Gradient. To guarantee the stability of the proposed G-PDE, this paper prove that G-PDE mitigates the problem of exploding and vanishing gradients and improves the trainingability of deep multilayer GNNs.

**Weaknesses:**

1.	How does the first-order and second-order implemented? The authors should describe it more detail.
2.	Additionally, what’s the connection between the second-order spike representation and the second-order graph-PDE?

**Questions:**

See weaknesses